# Use of Olive and Sunflower Oil Hydrogel Emulsions as Pork Fat Replacers in Goat Meat Burgers: Fat Reduction and Effects in Lipidic Quality

**DOI:** 10.3390/biom12101416

**Published:** 2022-10-03

**Authors:** Iasmin Ferreira, Lia Vasconcelos, Ana Leite, Carmen Botella-Martínez, Etelvina Pereira, Javier Mateo, Seyedalireza Kasaiyan, Alfredo Teixeira

**Affiliations:** 1Centro de Investigação de Montanha (CIMO), Instituto Politécnico de Bragança, Campus de Santa Apolónia, 5300-253 Bragança, Portugal; 2Laboratório para a Sustentabilidade e Tecnologia em Regiões de Montanha, Instituto Politécnico de Bragança, Campus de Santa Apolónia, 5300-253 Bragança, Portugal; 3IPOA Research Group, Centro de Investigación e Innovación Agroalimentaria y Agroambiental CIAGRO, Miguel Hernández University, 03312 Orihuela, Spain; 4Escola Superior Agrária. Instituto Politécnico de Bragança, Campus de Santa Apolónia, 5300-253 Bragança, Portugal; 5Departamento de Higiene y Tecnología de los Alimentos, Facultad de Veterinaria, Campus Vegazana s/n, 24007 León, Spain

**Keywords:** goat meat, olive oil, sunflower oil, hydrogel emulsion, burgers, healthy products

## Abstract

Diversified strategies to incorporate healthier lipids in processed meat products are being developed. Alternative fat sources to replace animal fat associated with the reduction of fat content are some of the methods used to obtain healthier meat products well recognized by consumers. In order to create a healthier product that can also be consumed in the Halal and Kosher consumer markets, an experimental study was developed to assess the effects of replacing the pork fat (4%) with the same amount of hydrogel emulsion incorporating olive oil or sunflower oil. Three burgers were randomly selected from each lot manufactured and analyzed in triplicate. Burgers were physicochemical analyzed for pH, water activity, composition, fatty acid profile, color, yield, texture, oxidative stability, and volatile compounds and compared according to the fat source. Burgers with hydrogel emulsions can be considered reduced-fat meat products with a healthier fatty acid profile than pork fat burgers. The use of hydrogel emulsions did not negatively affect the quality characteristics assessed in the product and improved the oxidative stability during the storage of cooked burgers. By the characteristics and formulations evaluated, the replacement of pork fat with olive oil hydrogel emulsion proved to be the most effective strategy for obtaining a healthier goat meat product.

## 1. Introduction

Recent social changes have led consumers to a greater awareness of healthy eating [1] and the food industry to continuously research and adopt different approaches to produce healthier foods, more attractive to consumers. Among those approaches is the food reformulation aimed to reduce energy density, salt, or saturated fat in meat products, or promote the presence of functional components or favorable nutrients [2]. In this regard, the meat industry is facing the challenge to replace animal fat with oils, the main purpose of this is to improve the fatty acid profile and reduce cholesterol and its oxidation derivatives [3,4,5,6,7,8,9,10]. The foremost difficult task is to avoid the negative effect of fat replacement on the technological and sensory quality traits to which animal fat positively contributes, such as flavor, mouthfeel, texture, and overall perception [8,9]. Facing this challenge, meat researchers are recently investigating the application of solid plant-based oil structured emulsions such as oleogels and hydrogels to replace animal fat [10,11,12].

Goat meat is less popular by far than pork, beef, or chicken in the occidental countries’ daily diet. However, goat meat is part of local food traditions, goat meat eaters represent a niche market, and goat meat consumption appeared to show a trend to increase in the last decade [13,14]. The potential of goat meat, from the nutritive point of view, relies mostly on being a healthier alternative to beef and pork, with lower fat content and interesting fatty acid (FA) profile [15]. The lipidic profile of lean goat meat is mostly composed of unsaturated FAs, as monounsaturated (MUFAs) represents approximately 45% and polyunsaturated (PUFAs) are 10% of total FAs [16]. In order to make it more appealing, different processing methods can be applied to this meat, such as curing, drying, or incorporation into patties, sausages, and burgers [17]. The processing of goat meat helps to avoid the disposal of meat considered not marketable as fresh meat, due to its unfavorable eating quality and adds value to those [18]. In the countries where the pork consumption is extensive the usual source of fat used to prepare fat-containing ruminant meat products, including goat meat products, such as sausages or patties is pork fat [19,20], due to its favorable technological properties and positive effects on their eating quality. However, its lipidic profile is not desirable for healthy food [3,4,5,6,7].

Previous studies have shown that several reformulation approaches, i.e., reducing fat content, using different fat sources, and adding cereal flours, in different goat meat products i.e., sausages, nuggets, or pâté, can result in better sensory and nutritional characteristics [8,21,22,23,24,25,26]. Following this research and considering that conventional burgers have a solid image among consumers not linked to healthiness, it can be hypothesized that the use of solid oil structured emulsion might improve the nutritional quality of goat burgers with no negative effect on their technological quality traits. This approach would result in new and healthier options for regular goat meat consumers, for the ones who have restricted diets when it comes to pork meat, or animal fat, and for consumers demanding Kosher and Halal meat products in the market [27]. The aim of this study is to evaluate the quality of goat meat burgers made with olive oil and sunflower oil hydrogel emulsions (HE), with a special focus on fatty acid profiles and volatile compounds.

## 2. Materials and Methods

### 2.1. Burger Formulations and Production

Three types of goat meat burgers were prepared, one using pork fat (GPF) and the other two using olive oil (GO) or sunflower oil (GSF) hydrogel emulsions as lipid sources (Table 1). The meat used for burgers consisted of fresh goat meat from the shoulder and loin of 5 years old Serrana Goats carcasses. Hydrogel emulsions used to produce the burgers were prepared following the guidelines set out by Barros et al. [28]: 56% of water, 37.3% of olive oil or sunflower oil, and 6.7% of Prosella^®^ (Laboratorios Amerex S.A.U., Madrid, Spain), a mixture composed of calcium sulfate, sodium alginate, wheat glucose, disodium diphosphate, and sodium ascorbate. The fatty acid profile of the pork fat used, coming from Bísaro pork belly, was C14:0, 1.3%; C16:0, 22.3%; C16:1, 2.1%; C18:0, 11.9%; C18:1n-9, 41.9%; C18:2n-6, 15.7%; C18:3n-3, 1.2%; and C20:1n-9, 0.78% [29]. The olive oil used was a Trás-os-Montes Protected Origin Designation (PDO) brand, containing: C16:0, 11.2%; C17:1n-8, 0.2%; C18:0, 3.3%; C18:1n-9, 75.2%; C18:2n-6, 7.7%; C20:0, 0.4%; C18:3n-3, 0.8%; C20:1n-9, 0.2%; and C22:0, 0.1% [29]. The sunflower oil used, from a commercial local brand, had the following nutritional allegation for lipid composition: SFA, 10%; MUFA, 28%; and PUFA, 53%.

Burgers were made at the Carcass and Meat Quality Laboratory at Agriculture School of Polytechnic Institute of Bragança (Portugal), according to the Portuguese Standard NP 588/2008 [30] and following the process from Teixeira et al. [7]. Briefly, the meat and the pork fat or hydrogel emulsions were grounded using a 30-mm sized plate in a butcher’s meat grinder, then salt and water were added, and the mixture was kneaded by hand for 10 min. A total of 800 g of each formulation was separately prepared and the mixtures were divided into 7 approximately 120 g balls, which were shaped into 5- to 6-cm diameter and 1.5- to 2.0-cm thick burgers and covered with food purpose cellophane plastic (Figure 1). The burgers were packaged in vacuum bags, sealed, and frozen at −18 °C for storage until further analysis. Before analysis, the burgers were thawed at cold temperature (4 °C) for 24 h and, after being removed from the packaging, each analysis was performed in triplicate (in three different burgers per treatment). Results were then summed up to achieve the average values.

### 2.2. Raw Burger Technological Quality Traits and Composition Analyses

Burger pH was measured according to the Portuguese Standard NP-ISO 3441/2008 [31], with a Crison 507 pH-meter equipped with a 52–32 puncture electrode (Crison Instruments, Barcelona, Spain). Water activity (a_w_) was determined according to AOAC [32] using a probe HigroPalmAw1 Rotronic 8303 (Bassersdorf, Switzerland). Color measurements were obtained as CIELab coordinates using a Lovibond RT Series—SP62 spectrophotometer (The Tintometer Limited, Wiltshire, UK), with the equipment using a range from 400 to 700 nm for the measurement taken in 10 nm intervals [33] (pp. 17–23). Each burger was evaluated three times on both surfaces for lightness (L*), redness (a*), and yellowness (b*), from which chroma (saturation index) and hue angle were calculated, according to the Equations (1) and (2), respectively.
(1)$$C^{*}=\sqrt{\left({a^{*}}^{2}+{b^{*}}^{2}\right)}$$
(2)$$h^{*}=\tan^{-1}\left(a^{*}/b^{*}\right)$$

The oxidation status of the raw burgers, assessed by the thiobarbituric acid reactive substances (TBARS), was determined by extraction of malondialdehyde (MDA) as stated in NP-ISO-3356/2009 with modifications. In the extraction procedure, approximately 2 g of sample were used and after homogenization with 20 mL of distilled water, TCA 25% was added and centrifuged at 12,000 rpm for 15 min. The results were expressed as mg of MDA/kg of the sample [34].

Moisture content was analyzed according to NP 1614/2002 [35]; briefly, approximately 3 g of sample were added with 5 mL of ethanol and heated at 70 °C until its complete evaporation; after, they were oven-dried at 103 °C ± 2 °C until constant weight. For ashes content, samples were incinerated in muffle at 550 °C, according to NP-ISO 1615/2002 [36]. Total lipidic content was extracted from a 25-g sample amount, according to Folch et al. [37]. Protein content was assessed in accordance with NP 1612/2006, using the Kjeldahl method [38]. Myoglobin was assessed as defined by Hornsey [39] using the reflectance of the exposed surface by spectroscopy with a Spectronic Unicam 20 Genesys at 512 nm and expressed as mg myoglobin/g fresh muscle. Collagen content was performed via hydroxyproline determination as described in NP 1987/2002 [40], and chloride content, expressed as sodium chloride as a mass percentage, was measured following the Portuguese Standard NP 1845/1982 [41]. Finally, the FA profile was determined from approximately 50 mg of the extracted fat. Transesterification was performed as described by Shehata et al. [42] with the modifications applied by Domínguez et al. [43] and detailed in Teixeira [29]. Results were expressed as g of FA/100 g of total FA. The FA profile quality was assessed by PUFA/SFA and n-6/n-3 ratios (Department of Health—London, UK, 1994) [44], and the indexes of atherogenicity (AI) and thrombogenicity (TI), calculated according to Ulbricht and Southgate Equations (3) and (4) [45]. The ratio of hypocholesterolemic and hypercholesterolemic fatty acids (h/H), also was determined by Equation (5) as stated by Santos-Silva, Bessa and Santos-Silva [46].
(3)$$AI=\frac{C12:0+4\;x\;C14:0+C16:0}{\sum \mathrm{MUFA}+\sum \mathrm{PUFA}}$$
(4)$$TI=\frac{C\;14:0+C16:0+C18:0}{0.5\times \sum \mathrm{MUFA}+0.5\times \sum \mathrm{PUFA}\;n-6+3\times \sum \mathrm{PUFA}\;n-3+\frac{\mathrm{PUFA}\;n-3}{\mathrm{PUFA}\;n-6}}$$
(5)$$\frac{h}{H}=\frac{C18:1n-9+\;C18:2n-6+C20:4n-6+C18:3n-3+\;C20;5-n3+C22:5n-3+C22:6n-3}{C14:0+C16:0}$$

### 2.3. Cooked Burger Analyses

Sets of three burgers per treatment and batch, were weighed, measured, and then cooked in a convection oven in the fan-forced and upper and lower heating mode at 200 °C for 20 min until the inside temperature reached 80 °C. Burgers were turned upside down once during cooking. After being temperate for 1 h at room temperature (15 °C), weight loss (WL) was evaluated by the weight difference between fresh and cooked burgers and expressed as a percentage (Equation (6)). Moreover, the dimensional changes due to cooking were determined by measuring the thickness increase (TI) and shrinkage, both expressed as a percentage and determined by Equations (7) and (8) [47].
(6)$$\%Weight\;loss=\frac{\mathrm{raw}\;\mathrm{weight}-\mathrm{cooked}\;\mathrm{weight}}{\mathrm{raw}\;\mathrm{weight}}$$
(7)$$\%Thickness\;increase=\frac{\mathrm{cooked}\;\mathrm{thickness}-\mathrm{raw}\;\mathrm{thickness}}{\mathrm{cooked}\;\mathrm{thickness}}$$
(8)$$\%Shrinkage=\frac{\mathrm{raw}\;\mathrm{diameter}-\mathrm{cooked}\;\mathrm{diameter}}{\mathrm{raw}\;\mathrm{diameter}}$$

Burgers were then cut into two halves. One-half was used for texture profile analysis (TPA). This half was refrigerated for 24 h at 4 °C and then three 1-cm-sided cubes were obtained from the central zone. TPA was analyzed with a texture analyzer TA.XT2i (Stable Micro Systems, Godalming, Surrey, UK) is equipped with a 5-cm diameter cylindrical probe set to run at 1 mm/s and a compression depth of 80% of the thickness of the sample. Mean values for hardness, elasticity, and cohesiveness were obtained from the force–time curves, and chewiness was calculated as the product of hardness, cohesiveness, and springiness. The other half was used to analyze TBARS and headspace volatile contents. Previously, this half-burger was divided into two quarters. One quarter was homogenized with a food processor and analyzed (recently cooked burgers). The other quarter was wrapped with (high oxygen permeability) polyvinyl chloride cling film and analyzed after three days of aerobic refrigerated storage (4 °C; 72 h refrigerated stored cooked burgers).

TBARS contents were determined following de above-mentioned procedure. Volatile compounds were analyzed by gas chromatography-mass spectrometry using the solid-phase micro-extraction (SPME) technique following the procedure described by Carballo et al. [48] with the following modifications. Briefly, the extraction of volatiles was carried out with a CTC Pal automated system (Agilent Technologies; Santa Clara, CA, USA) equipped with an automatic SPME injection device using a 30-min 250-°C conditioned 75 mm Carboxen/polydimethylsiloxane-1-cm-coated fused silica fiber, from 3 g of grounded burger sample placed into a 20-mL screw cap vials, which were previously incubated at 45 °C during 20 min, for a 40 min exposition period at 40 °C. A 60 m × 0.25 mm sized, 0.25 mm film thickness DB-5MS column (J&W Scientific, Folson, CA, USA) and helium (1 mL/min) were used for separation with the temperature being programmed at 35 °C (1 min), 35 °C to 50 °C (10 °C/min), 50 °C to 200 °C (4 °C/min), 200 °C to 250 °C (50 °C/min) and 250 °C (11 min). The mass spectrometer transfer line and ion source temperatures were 240 °C and 260 °C, respectively, and the detector operated in electron impact mode (70 eV) scanning from 40 m/z to 350 m/z. Identification was carried out by comparing the mass spectra with those contained in the NIST/EPA/NIH-98 mass spectral database and the linear retention indexes, experimentally calculated using a series of n-alkanes with those reported in the literature [49]. The concentrations of the identified compounds were expressed as area units (AU) × 10^6^.

### 2.4. Statistical Analysis

To analyze the differences between burger formulations, a standard least square model was fitted using the statistical package JMP^®^ Pro 16.0.0, 2021, SAS Institute Inc. © (Cary, NC, USA) . When the ANOVA was significant, the predicted means were ranked based on pair-wised least significant differences and compared using Tukey’s HSD test for *p* < 0.05, *p* < 0.01, or *p* < 0.001 significance levels. A discriminant analysis was performed using the linear, common covariance, and stepwise variable selection methods (PROC DISCRIM, SAS). The efficiency of the discriminant model was assessed by the test of Wilks’ lambda value. Results were analyzed in terms of absolute assigned individuals to a pre-assigned group and the variance explained by each canonical resemblance as well as by the analysis of the scoring coefficients.

## 3. Results and Discussion

### 3.1. Raw Burgers Quality Traits and Composition

The values of the quality traits and composition of the burgers prepared are shown in Table 2. Although the GO-pH value was significantly lower than the others, the mean pH values between treatments differed by less than 0.05 units. HE contained 56% of water, which is a considerably higher amount than the water content of pork backfat and, therefore, it would have been expected that a_w_ of fat replaced burgers were higher than that of GPF ones. However, the a_w_ of GPF burgers was the highest among the formulations tested. The a_w_-reducing effect of hydrogel could be attributed to its components, namely mineral salts (CaSO_4_, Na_2_H_2_P_2_O_7_), glucose, and Na^+^ from the alginate and ascorbate salts incorporated into the HE burgers. The OH-groups of glucose might have played a significant role in that effect [48] considering that the amounts of ash or Na^+^ originated from the hydrogels were not high enough to cause a significant increase in ash and NaCl contents of GO and GSF burgers as regards to GPF burgers (Table 2).

Referring to color parameters, the only CIELab coordinate affected by the fat replacement was b* (yellowness), finding GO to have the highest value. This difference can be caused by the presence of the olive oil HE as reported in other studies, implying that the color of the olive oil used had an impact on the product [7,50]. Lipidic oxidation did not present statistical differences between formulations. It is important to assure the lipidic stability of the fat-replaced burgers given that the high level of polyunsaturation of vegetable oil FA could be a factor to make the fat-replaced burgers highly susceptible to oxidation, thus causing major alterations in their sensory characteristics; however, on the other hand, vegetal oils usually contain higher antioxidant capacity than animal fats [51], which contribute to preventing lipid oxidation. In this case, the lipid source was not a significant factor in the lipid oxidation of burgers and TBARS values were below the minimum limit reported in order to detect rancidity flavors [52,53,54].

Concerning the proximate composition, in agreement with burger formulations (Table 1) and the composition of the ingredients used, there were no statistical differences between burgers when it comes to moisture, ashes, protein, and NaCl contents. In contrast, the fat content of GPF burgers was significantly higher than that of HE burgers, explained by the hydrogel composition since the oils represented 37.3% of its formulation. Regarding the collagen, GPF burgers (0.34%) showed the highest content, which was significantly higher than the burger with sunflower oil HE; this result can be explained by the presence of collagen in pork fat [55]. Despite the variation in moisture content among batches, probably due to the water incorporated into the burger mix through the addition of HE (approximately 2.3%) resulting in higher moisture mean values, the moisture differences were not significant.

The lipid sources caused major changes in the FA profile (Table 3). As expected, and shown in previous studies where oils were added to meat products, the FA content was impacted accordingly to the oil FA profiles [28,50,56,57,58]. As for the major FA, oleic acid (C18:1n-9), its content was statistically different for all formulations used and as presumed, GO presented the highest value, followed by GPF and GSF. For the content of linoleic acid (C18:2n-6) the burger that differed from the others was GSF, with superior content. GPF burgers, with the highest SFA content, had similar values for C16:0, C18:0, C20:1n-9, and C20:2n-6 than the ones reported by Teixeira et al. [7] and by Özer and Çelegen [59]. Burgers with olive oil (GO) had the highest contents of C18:1n-9, C18:3n-3, and C20:3n-3, which improved the levels of MUFAs and n-3. GSF burgers showed a higher percentage of C18:2n-6, as said before, and of C21:0, C24:0 and C22:6n-3. Thus, PUFA and n-6 percentages were also the highest in GSF burgers. No difference was found in AI. However, the TI of fat-replaced burgers showed a reduction compared to GPF, with no differences being detected between GO and GFS. The h/H values were also different between GFS and the other two burger groups, with the former having the lowest value. To sum up, the use of HE improved the goat-meat burger lipid-healthiness by reducing the SFA, TI, and h/H. Furthermore, among the HE burgers, GO seems healthier as a result of the higher content of n-3 percentage and lower n-6/n-3 ratio.

### 3.2. Cooked Burgers Analysis

The weight losses and shape changes of burgers during cooking and the TPA results in the cooked burgers are shown in Table 4. Weight loss percentage presented significant differences, with GSF being the treatment with the highest value. Previous studies using similar agents to replace animal fat in cooked burgers have found that hydrogel effects on weight loss were variable depending on lipid source, emulsion, and gelling or structuring agents [56,60,61,62]. In this study, compared with the GPF, the performance of HE burgers in yield was either similar (GO) or higher (GSF). With regard to GO and GSF comparison, since the only difference in composition between the two HE used was the oil source, this would have been responsible for the differences. The mechanism to explain this effect deserves further research. Regarding shrinkage and TI, there were no statistical differences between control burgers and HE ones.

The results of TPA analysis showed minor effects since only cohesiveness was significant, although slightly, influenced by the use of HE in goat burgers. The higher cohesiveness in the HE burgers suggests that the structuring agents, alginate, in this case, would have contributed to increasing the intermolecular links between meat particles without a significant effect on the other TPA characteristics. Different and variable results have been reported in other studies on the effect of hydrogels on reduced-fat burger TPA, i.e., increasing or decreasing the hardness and chewiness, affecting or not cohesiveness or elasticity [28,50,56,59,63,64,65]. In each of these studies a different HE was used, either formed with different oils or structuring agents, thus the discrepancy within studies can be justified by the many factors of formulation and making process with potential influence in burger TPA results, not only related to the hydrogel but also meat and fat types and amounts, salt amount, degree of meat comminution, the intensity of mechanic treatment for mixing, burger shape or cooking conditions.

It was observed that the treatment did not show any statistically significant differences in TBARS immediately after cooking. In other words, the addition of HE did not affect the MDA values in recently cooked burgers despite having a higher amount of polyunsaturated fatty acids and, therefore, more susceptibility of lipid to oxidation in the substituted samples might have been expected, similar results were found by Barros et al. [28]. On the other hand, lipid oxidation after 72 h of storage, as seen in Figure 2, considerably increased, from 8.5 times for GO burgers to 14 times for GPF burgers, and statistical differences in TBARS of refrigerated stored burgers were detected. The increase in oxidation during storage time was slowed down in the samples with HE.

Table 5 shows the volatile compound profile in the headspace of burgers at two different times: immediately after cooking and after a 72-h aerobic refrigerated storage. The quantified compounds were mostly straight-chain aldehydes which together with the also quantified octanedione and furan-2-pentyl (both coeluting with decane) are considered to be relevant odor-active compounds derived from lipid degradation in cooked meat [66,67]. Moreover, in second place, straight-chain hydrocarbons, mainly pentane and octane, were also abundant. These hydrocarbons, which would also have originated from lipids, are abundant in cooked meat, goat meat included [68,69]. Although the straight-chain hydrocarbons can contribute to a pleasant meat flavor [68], their effect on cooked meat flavor is of little relevance as compared with aldehydes, due to higher odor thresholds [70].

Among the aldehydes detected, as expected for cooked meat in general, hexanal dominated. This study shows that the formulation (lipid source) of burgers had a significant effect on the levels of hexanal, and also in those of nonanal and the sum of aldehydes. Moreover, storage time resulted in a significant increase in the amounts of all aldehydes. Actually, the effect of treatment on the volatile aldehyde contents was only evidenced in burgers after 72 h of refrigerated storage, i.e., the effect was not observed in the recently cooked burgers. The increase in aldehydes observed in burgers during storage indicates that the process of lipid oxidation progresses in all three types of burgers during this step. It is well known that lipid oxidation in cooked meat during storage and eventually reheating elicits the formation of carbonyl compounds and the subsequent development of the characteristic off-flavor named warmed-over-flavor (WOF) [71].

In our study, the significant interaction effect observed between burger type and storage time shows that the rate of lipid oxidation during cooked burger storage varied among burger types. This variation in the lipid oxidation rate was also observed in the TBARS results (Figure 2). In agreement with TBARS levels, hexanal contents, also considered an index for lipid oxidation [71], were higher in stored GPF burgers than in stored HE burgers, suggesting, both index (TBARS and hexanal contents), a higher oxidation rate in the GPF lipids during storage, which means lower lipid stability when using pork fat. However, the pattern followed by nonanal was completely different from that of hexanal. Results showed that nonanal content was higher in GO burgers and lower in GSF, with GPF being in an intermediate position. The variability in the patterns showed by the different aldehydes between burger types can be attributed to differences in the lipid source FA profile. FA are the main precursors of aldehydes, and the degradation of different FA produces variability in the amounts of the formed aldehydes. Thus, the oxidation pathway of oleic acid is characterized by a higher formation of nonanal, and also octanal, than that of other unsaturated fatty acids such as linoleic acid [66,72]. Consequently, as observed in this study, Rodríguez-Carpena et al. [72] also found more nonanal (and also octanal) and a lower ratio of hexanal/nonanal in burgers formulated with olive oil than in burgers formulated with pork fat or sunflower oil, being the sunflower-containing burgers those with highest hexanal/nonanal ratio.

Differences due to different FA profiles in hexanal and nonanal, and eventually in other FA-derived odor-active volatile compounds presumably generated in the burgers during storage, although not detected with the method used due to an insufficient concentration in the headspace, such as hept-(E)-2-enal, oct-1-en-3-ol, octane-2,3-dione, and alka-2,4-dienals, would have affected the stored burger flavors. Specifically, these differences would have affected the intensity and unpleasantness of the WOF developed [63]. This is because all the above-mentioned compounds present very low odor thresholds. Previous studies have suggested that higher content of oleic acid and its oxidation-derived volatile compounds in burgers might result in better flavor acceptability [72]. The effect of lipid source on burger flavor after storage needs to be further clarified by means of sensory analysis.

Regarding the straight-chain hydrocarbons, the lipid source showed a significant effect on octane concentration which was present at the highest amount in recently cooked GO burgers. In agreement, octane has been found to be a major degradation compound formed from oleate during heating [73]. Most of the hydrocarbons detected were not significantly affected by burger storage. Only hexane and octane were affected, and their amounts were lower in stored burgers. These findings suggest that the straight-chain hydrocarbon presence in burger headspace was more linked to burger processing and cooking than to oxidation during storage.

By performing a stepwise discriminant analysis, it is possible to identify and correctly classify 100% of the burgers using only two variables corresponding to the FA, C20:2n-6 and C18:1n-9, and the model is highly significant (*p* < 0.0001 *; Wilks Lambda value: 0.00475) which is possible to see in Table 6.

The scatter plot of the two canonical variables of the three burger groups was discriminated with great accuracy with a total of 100% of variance explained, 78.95%, and 21.05% for canonical functions 1 and 2, respectively (Figure 3). The first canonical function discriminates between burgers made with olive oil appearing on the positive side of the axis and the burgers made with pork backfat and sunflower oil on the negative side of the axis.

## 4. Conclusions

The replacement of pork fat with vegetable oil HE in goat meat burgers was performed successfully in this study. Evaluating the results obtained, it is possible to affirm that the use of olive or sunflower oil HE in the burgers allows us to obtain reduced-fat burgers and that the fat replacement did not cause major changes in its quality either as raw or cooked burgers. Concerning lipidic profile, the changes were significant and in accordance with the oils used (olive and sunflower oil), lipidic healthiness was improved. HE affected the TBARS and volatile compounds of the burgers’ headspace after 72 h post cooking, suggesting higher oxidative stability during cooked burger storage. The discriminant analysis performed, identified correctly all formulations using only two FA (C18:1n-9 and C20:2n-6), indicating that besides their similar behavior through other analyses, they have a differentiating trait. In addition to that and considering the results obtained, it is possible to affirm that among the vegetable oils used, olive oil presented better characteristics when it comes to lipidic healthiness. Moreover, GO burger flavor, due to the release of higher levels of specific volatile compounds from oleic acid, might be more positively perceived by consumers, although it can only be presumed as a sensory analysis is needed to confirm such affirmation.

## Figures and Tables

**Figure 1 biomolecules-12-01416-f001:**
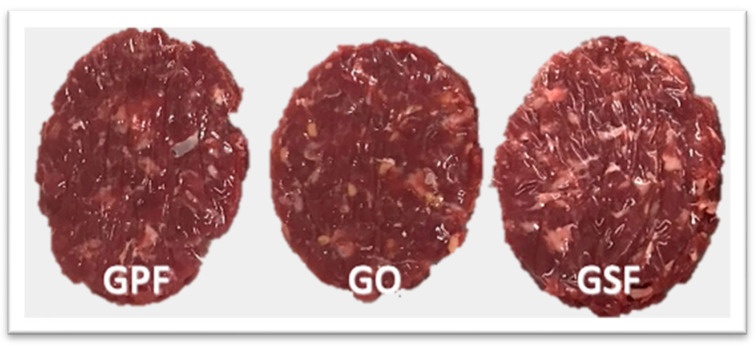
Fresh burgers covered with oval clear plastic cellophane discs.

**Figure 2 biomolecules-12-01416-f002:**
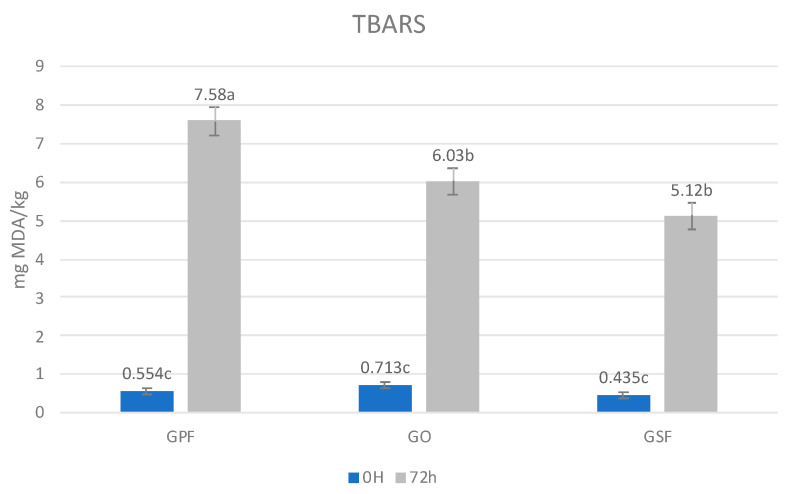
TBARS (mg MDA/kg of sample) of cooked goat meat burgers at 0 h and 72 h (Standard Error of Mean ± 0.351, Significance: 0.009, a–c mean values shown in bars not followed by a common letter differ significantly (*p* < 0.05; Tukey test).

**Figure 3 biomolecules-12-01416-f003:**
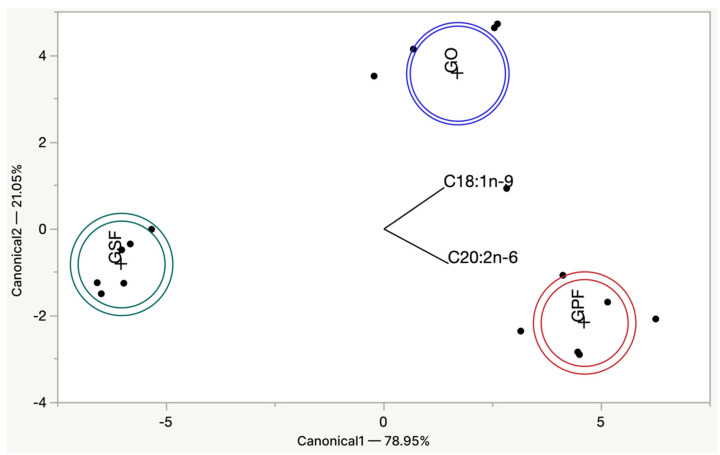
Scatter plot of the first two canonical variables of the three burger groups.

**Table 1 biomolecules-12-01416-t001:** Formulation of goat meat burgers expressed in percentage.

Ingredients (%)	GPF	GO	GSF
Goat meat	87.9	87.9	87.9
Pork fat	4	-	-
Olive oil hydrogel	-	4	-
Sunflower oil hydrogel	-	-	4
NaCl	1.1	1.1	1.1
H_2_O	7	7	7

GPF: Pork fat; GO: Olive oil hydrogel emulsion; GSF: Sunflower oil hydrogel emulsion.

**Table 2 biomolecules-12-01416-t002:** Technological and chemical properties of goat meat burgers with pork fat replacement by HE.

Parameter	GPF		GO		GSF		SEM	SIG
pH	5.91	b	5.89	a	5.92	b	0.009	0.017
a_w_	0.982	a	0.968	a, b	0.967	b	0.005	0.024
L*	45.45		46.28		44.55		0.628	0.196
a*	18.64		19.86		18.23		0.811	0.362
b*	15.41	b	16.81	a	15.03	b	0.251	0.007
h*	39.63		40.27		39.51		0.816	0.766
C*	24.19		26.02		23.63		0.775	0.140
TBARS (mg MDA/kg)	0.474		0.386		0.511		0.069	0.431
Moisture (%)	73.57		76.71		75.61		1.243	0.216
Ashes (%)	1.93		1.96		2.03		0.078	0.572
Fat (%)	5.46	a	3.79	b	3.26	b	0.069	0.030
Protein (%)	18.10		17.82		17.68		0.265	0.647
Pigments (%)	2.86		3.18		2.88		0.199	0.707
Collagen (%)	0.34	a	0.29	a, b	0.21	b	0.038	0.019
Chlorides (%)	1.15		1.10		1.29		0.069	0.176

SEM: Standard Error of Mean; SIG: Significance value; a, b mean values in the same row not followed by a common letter differ significantly (*p* < 0.05; Tukey test).

**Table 3 biomolecules-12-01416-t003:** Lipidic profile of goat meat burgers with fat replacement.

Fatty Acids (%)	GPF		GO		GSF		SEM	SIG
C14:0	2.016		2.104		2.669		0.501	0.568
C14:1	0.094		0.121		0.118		0.040	0.754
C15:0	0.202		0.285		0.256		0.031	0.169
C16:0	23.762	a	19.529	b	17.911	a	0.683	<0.0001
C16:1n-7	2.171		1.899		1.904		0.336	0.784
C17:1n-7	0.427		0.506		0.359		0.056	0.199
C18:0	12.093	a	9.613	a,b	9.334	b	0.935	0.022
9t-C18:1	1.164		1.396		1.285		0.197	0.691
C18:1n-9	46.417	b	53.632	a	38.872	c	0.877	<0.0001
9t,12t-C18:2	0.114		0.150		0.143		0.028	0.563
C18:2n-6	7.643	b	6.921	b	24.070	a	1.150	<0.0001
C20:0	0.152		0.176		0.088		0.032	0.143
C18:3n-6	0.024	a	0.017	a,b	0.004	b	0.007	0.015
C20:1n-9	0.689	a	0.195	b	0.080	b	0.041	<0.0001
C18:3n-3	0.430	a	0.526	a	0.238	b	0.041	0.001
C21:0	0.354		0.498		0.468		0.113	0.974
C20:2n-6	0.291	a	0.040	b	0.010	b	0.018	<0.0001
C22:0	0.079	b	0.074	b	0.256	b	0.038	0.004
C20:3n-6	0.076	a	0.042	a,b	0.029	b	0.014	0.008
C20:3n-3	0.068	a, b	0.167	a	0.029	b	0.041	0.025
C20:4n-6	0.851		1.093		0.916		0.087	0.144
C24:0	ND	b	0.014	a,b	0.058	a	0.017	0.045
C24:1n-9	0.099	a	0.089	a	0.040	b	0.016	0.049
C22:6n-3	0.046	b	0.082	a,b	0.116	a	0.022	0.031
∑SFA	39.28	a	33.00	b	31.68	b	0.883	<0.0001
∑MUFA	51.12	b	57.90	a	42.70	c	1.109	<0.0001
∑PUFA	9.59	b	9.10	b	25.62	a	1.170	<0.0001
∑UFA/∑SFA	1.55	b	2.04	a	2.17	a	0.079	<0.0001
∑n-6	8.89	b	8.11	b	25.03	a	1.153	<0.0001
∑n-3	0.59	b	0.83	a	0.52	b	0.078	0.027
∑n-6/∑n-3	15.96	b	10.81	b	50.48	a	2.914	<0.0001
AI	0.53		0.42		0.42		0.041	0.652
TI	1.19	a	0.89	b	0.85	b	0.040	<0.0001
h/H	2.15	b	2.95	a	3.25	a	0.213	0.004

ND: not detected; SFA: saturated fatty acids; MUFA: monounsaturated fatty acids; PUFA: polyunsaturated fatty acids; UFA: unsaturated fatty acids; AI: atherogenicity index; TI: thrombogenicity index; SEM: Standard Error of the Mean; SIG: Significance value; a–c mean values in the same row not followed by a common letter differ significantly (*p* < 0.05; Tukey test).

**Table 4 biomolecules-12-01416-t004:** Weight loss and change of shape due to cooking and TPA of goat meat cooked burgers with fat replacement.

Parameter	GPF		GO		GSF		SEM	SIG
WL (%)	64.30	b	64.60	b	70.75	a	0.961	0.029
Shrinkage (%)	27.26		22.71		22.75		1.505	0.398
TI (%)	20.22		27.59		30.99		4.882	0.191
Max Force (N)	101.80		113.33		108.83		4.129	0.283
Cohesivity	0.337	b	0.366	a	0.373	a	0.005	0.030
Elasticity	0.640		0.591		0.587		0.022	0.314
Chewiness	22.19		24.52		23.81		0.962	0.346

WL: Weight Loss; TI: Thickness increase; SEM: Standard Error of the Mean; SIG: Significance value; a, b mean values in the same row not followed by a common letter differ significantly (*p* < 0.05; Tukey test).

**Table 5 biomolecules-12-01416-t005:** Volatile compound in HS of goat meat burger with fat replacement by HE.

Compound	GPF				GO				GSF				SEM	SIG		
	0H		72 h		0H		72 h		0H		72 h			FS	T	FS × T
Pentane	477.0		778.6		500.6		476.4		858.2		1336.8		309.30	0.200	0.356	0.725
Hexane	50.8	a	20.65	b	46.5	a	28.05	b	51.8	a	36.6	b	9.44	0.654	0.033	0.720
Heptane	158.3		47.8		162.2		103.5		245.8		125.0		67.98	0.511	0.132	0.889
Octane	405	c	138.6	c	1458.4	a	475.0	c	935.6	b	433.0	c	125.52	0.004	0.001	0.071
Octene	8.9		25.6		20.4		9.3		22.4		17.6		9.23	0.860	0.097	0.353
Heptane, 2,2,4,6,6-pentamethyl	98.2		166.2		143.2		77.9		76.2		81.1		50.10	0.589	0.952	0.460
Sum of hydrocarbons	1198.8		1177.4		2331.2		1170.2		2189.8		2030.1		465.36	0.217	0.284	0.458
Ethanal	18.5	b, c	25.3	a, b	12.8	c	29.8	a	14.2	c	26.8	a, b	2.42	0.854	0.001	0.183
Pentanal	181.2	a, b	477.2	a	167.4	a, b	416.6	a, b	151.9	b	407.4	a, b	91.45	0.857	0.012	0.962
Hexanal	4038.8	d	11,480.2	a	2255.3	e	9728.2	b	3057.2	d, e	7593.2	c	338.84	0.001	<0.0001	0.007
Heptanal	260.7	b, c	487.5	a, b	230.8	b, c	690.7	a	91.8	c	435.0	a, b	88.10	0.160	0.003	0.464
Octanal	5.75	b	90.2	a, b	ND		140.2	a	ND		36.1	a, b	33.02	0.352	0.018	0.354
Nonanal	ND		124.8	b	ND		257.4	a	ND		96.5	b	20.29	0.016	<0.0001	0.016
Sum of aldehydes	4504.9	c	12,685.2	a	2666.3	d	11,262.8	a	3315.0	c, d	8595.1	b	505.41	0.006	<0.0001	0.033
Octanedione + dodecane + 2-pentyl-furan	183.0	b	2512.5	a,b	42.7	b	3010.8	a	41.4	b	741.0	a,b	748.36	0.332	0.017	0.359
Carbon disulfide	63.2		94.6		106.0		62.2		105.5		139.0		48.68	0.645	0.865	0.682

SEM: Standard Error of the Mean; SIG: Significance value; a–e mean values in the same row not followed by a common letter differ significantly (*p* < 0.05; Tukey test). ND: not detected.

**Table 6 biomolecules-12-01416-t006:** Values Prob > F from discriminant analysis of goat meat burgers with fat replacement.

Fatty Acid	F Ratio	Prob > F
C4:0	0.094	0.911
C8:0	0.451	0.647
C10:0	2.299	0.143
C11:0	2.247	0.148
C12:0	0.658	0.536
C14:0	0.960	0.410
C14:1	0.554	0.589
C15:0	0.620	0.554
C15:1	0.467	0.638
C16:0	1.059	0.377
C16:1n-7	0.939	0.418
C17:0	0.831	0.459
C17:1n-7	0.770	0.485
C18:0	1.551	0.252
9t-C18:1	0.556	0.588
**C18:1n-9**	**91.503**	**0.000**
9t.12t-C18:2	0.341	0.718
C18:2n-6	1.722	0.220
C20:0	1.080	0.370
C18:3n-6	1.437	0.276
C20:1n-9	3.138	0.080
C18:3n-3	2.825	0.099
C21:0	0.669	0.530
**C20:2n-6**	**106.620**	**0.000**
C22:0	0.773	0.483
C20:3n-6	1.036	0.385
C22:1n-9	0.025	0.975
C20:3n-3	2.011	0.176
C23:0	3.237	0.075
C20:4n-6	1.869	0.196
C22:2n-6	1.755	0.214

## Data Availability

Not applicable.

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
