# Peer review of "Use of Olive and Sunflower Oil Hydrogel Emulsions as Pork Fat Replacers in Goat Meat Burgers: Fat Reduction and Effects in Lipidic Quality"

_biomolecules, 2022, doi:10.3390/biom12101416_

Round 1

Reviewer 1 Report

This paper would like to use oleogel to replace pork fat in burgers. The experiments were designed well. However, it does not make sense to replace pork fat with a goat meat burger. I would suggest redesigning this paper. Please either develop the goat meat burger with oleogel or replace pork fat without mentioning goat meat. Therefore, I would suggest rejecting this paper, unless the authors can justify why replace pork fat with a goat meat burger.

Major issue

The replacement of pork fat ….. obtaining a goat meat product. This sentence is confusing. Is the goat meat product originally have pork fat in it?

Line 55 “pork fat is the main source of fat used in goat meat products.” None of the references from 3 to 7 can support this narrative. Please justify. Is pork fat allowed in the halal market? If not, how does the halal market deal with goat meat? What is “the oil” they use? What is the advantage of oleogel compared to “the oil”?

If the author can not prove “pork fat is the main source of fat used in goat meat products.”, the title, abstract, and introduction of this paper are meaningless.

The introduction needs to ask the following questions: 1, why don’t use plant oils directly; 2, What kind of oil has been added to goat meat on the halal market? 3, Compare that oil with oleogel rather than the pork oil.

Please list all the ANOVA results in the figures and tables.

Minor issue

Line 54- 55 Add value to those

Line 79 “from”

Please look through the grammar issues.

Author Response

Dear reviewer,

We thank you for the time spent reviewing the article. We regret the fact that you do not agree with the publication of this article. All modifications were made following the reviewer's suggestions and comments, and responses to their comments are also attached. Thanks to their recommendations, significant modifications were made throughout the manuscript.

Thank you for your attention.

__________________________________________________________________________________________________

This paper would like to use oleogel to replace pork fat in burgers. The experiments were designed well. However, it does not make sense to replace pork fat with a goat meat burger. I would suggest redesigning this paper. Please either develop the goat meat burger with oleogel or replace pork fat without mentioning goat meat. Therefore, I would suggest rejecting this paper, unless the authors can justify why replace pork fat with a goat meat burger.

Response: The main goal of the work is to replace pork fat in goat meat burgers. The work was created in order to improve the lipidic profile of an already created alternative product (goat burger besides beef or pork burger), normally the fat source for meat products is pork fat, having this in mind the experimental design of the work was proposed. Also the use of goat meat, already in other published research, reaffirm the importance to have an halal as well as kosher meat product and give added value to goat meat.

Major issue

The replacement of pork fat ….. obtaining a goat meat product. This sentence is confusing. Is the goat meat product originally have pork fat in it?

Response: In some situations, the back pork fat is used as an emulsifier and flavored agent to burgers independently the specie of the meat used. When talking about goat meat products, the aim is to mention a scope of products made of goat meat, that can use the same strategy (oleogels) to make it healthier and suitable for markets with restrictions towards pork (meat and fat) consumption.

Line 55 “pork fat is the main source of fat used in goat meat products.” None of the references from 3 to 7 can support this narrative. Please justify. Is pork fat allowed in the halal market? If not, how does the halal market deal with goat meat? What is “the oil” they use? What is the advantage of oleogel compared to “the oil”?

Response: Alleging that pork fat is the main source of fat in goat meat products, doesn’t reduce it to the only one used, as stated by the references: 13, 16, 17, 18, 19, 20, 21, 24, 25. The consumption of goat meat is mostly from fresh meat and not the processed products, the “development” of this product can be seen as an addition to common and halal markets. The use of oleogel instead of the vegetable is justified by the structural difference between fat and oil, applying vegetable oils directly to the burgers would cause major alterations in the burger texture, mouthfeel and assemblance. Using oleogels allows the addition of vegetable oils with a similar structure to the pork fat, as is possible to see in the figure 1 (white dots in GO and GSF) and doesn’t compromise the texture and visual aspect.

If the author can not prove “pork fat is the main source of fat used in goat meat products.”, the title, abstract, and introduction of this paper are meaningless.

Response: The introduction needs to ask the following questions: 1, why don’t use plant oils directly; 2, What kind of oil has been added to goat meat on the halal market? 3, Compare that oil with oleogel rather than the pork oil.

Please list all the ANOVA results in the figures and tables.

Minor issue

Line 54- 55 Add value to those - Changed

Line 79 “from” - Changed

Please look through the grammar issues.

Reviewer 2 Report

This manuscript evaluates the potential of olive and sunflower oil oleogels as potential pork fat replacers in goat meat burgers. The article is well-written and pleasant to read. The topic is of interest, in particular in the current context of developing plant-based products.

Major comments:

The report and interpretation of the fatty acid composition is a bit misleading.

- In the material and methods, the authors mentioned concentrations of fatty expressed in g of FA/100 of fat, while the results seems to be expressed as percentage of total fatty acids. Although, in most food, fatty acids constitute the most abundant part of fat, the term “fat” include all the lipids and not only the fatty acids. If the authors indeed looked at the concentration (and not percent composition), more details about the standards used need to be included in the material and methods.

- The calculation of the AI, TI and h/H seems to be based on the percent FA composition of the oil/fat fraction. To evaluate the AI, TI, and h/H of the burgers, it would be more accurate to consider the concentration in the burgers as the total fat content vary between them.

- Along the lines of the previous comment, for the interpretation of the lipid oxidation results, it would be better to consider the concentration of unsaturated fatty acids in the burgers, rather than in the oil. The percentage of unsaturated fatty acids might indeed be higher in the vegetable oil; however, as the amount of fat in these burgers was lower than the one with the pork fat, it is possible that the end concentration in the final product is lower.

Regarding the flavour profiles, the authors mostly focus on volatiles related to lipid oxidation. In cooked meat, Maillard reaction products also played an important role in the flavour of the product. Did the authors also consider looking at such products?  

The added value of the PCA analysis based on FA profile is not so clear. It would be important to highlight it better in the discussion of the results.

Other comments:

- Line 79: “from” is repeated twice.

- Section 2.1: The fatty acid profile of pork fat and olive oil is provided, while only the total SFA, MUFA and PUFA content is reported for sunflower oil. It would be nicer to be consistent and either report the fatty acid profile for all or only the total SFA/MUFA/PUFA for all.

- Lines 196-197: It is a bit misleading to first state that the pH values were similar for all treatments and then, mentioned that they were different.

- Line 199: It would be nice to mention here the water content of pork backfat between brackets to give a clearer idea on how much lower it is compared to oleogels.

- Lines 201-203: The authors mentioned that the variations in water activity cannot be attributed to the oleogel mineral salts. Do they have an hypothesis on what could explain their outcomes? Also, although the water content in oleogels is higher than in pork fat, the moisture content in the burgers are similar. Can the authors comment on that?

- Line 212: Could the authors be more specific in why the use of olive oil changes the yellowness, for example by the presence of specific pigments in olive oil?

- Lines 217-218: Did the authors consider measuring the antioxidant capacity of their fat to confirm their hypothesis?

- Lines 226-227: Although the presence of collagen in pork fat could explain the higher content of collagen in the burger with pork fat compared to sunflower oil, it is unclear then why no such difference was observed between the burger with pork fat and the one with olive oil oleogel.

- Line 229: major changes in the FA profile instead of major changes at the FA profile.

- Lines 260-262: It is unclear what the authors mean by cooking yield and where the reader can find the value related to the cooking yield.

- Reporting the standard deviation (SD) for each burger rather than the overall SEM in tables and figures will provide the reader a better estimation of the variability of the results and help with their interpretation.

- Line 287: The term “initial time” is confusing, maybe reformulate to refer to the time just after cooking.

- Line 293: “was slower” instead of “was more slowed down”

- Tables: It would be nice to precise in the footnotes that the values presented are means and that they were obtained on triplicates with a sentence such as “Values are means (n=3)”

- Figure 2: It would be good to precise in the caption what the bars and error bars mean, i.e. Means +/- SD? Also, this figure seems to have a light grey outline that the authors might want to remove.

- Line 355: major instead of mayor

- Table 6: The first column is titled “Column”, another name such as “Fatty acid” would be more appropriate. Also, in this column, it would make more sense to bold the fatty acid names that were significantly different (i.e. as in the F ratio and Prob>F columns) rather than all the other ones.

Author Response

Dear reviewer,

All modifications were made following the reviewer's suggestions and comments, and responses to their comments are also attached. Thanks to their recommendations, significant modifications were made throughout the manuscript.

Thank you for your attention.

____________________________________________________________________________

Major comments:

The report and interpretation of the fatty acid composition is a bit misleading.

- In the material and methods, the authors mentioned concentrations of fatty expressed in g of FA/100 of fat, while the results seems to be expressed as percentage of total fatty acids. Although, in most food, fatty acids constitute the most abundant part of fat, the term “fat” include all the lipids and not only the fatty acids. If the authors indeed looked at the concentration (and not percent composition), more details about the standards used need to be included in the material and methods.

Response: Indeed, there was a mistake about the units, the fatty acids were expressed as percentage of total fatty acids, the change was made in the methods session.

- The calculation of the AI, TI and h/H seems to be based on the percent FA composition of the oil/fat fraction. To evaluate the AI, TI, and h/H of the burgers, it would be more accurate to consider the concentration in the burgers as the total fat content vary between them.

Response: Due to the percentages of fat of burgers were calculated and provided, the reader can estimate the significance of those indexes in the context of the burger total fat content. Moreover, our main aim is a compare the quality of the different burger fat. Other studies, such as that by Orkusz (2021). Nutrients https://www.mdpi.com/2072-6643/13/4/1207 also calculate the indexes based on the fatty acid profile

- Along the lines of the previous comment, for the interpretation of the lipid oxidation results, it would be better to consider the concentration of unsaturated fatty acids in the burgers, rather than in the oil. The percentage of unsaturated fatty acids might indeed be higher in the vegetable oil; however, as the amount of fat in these burgers was lower than the one with the pork fat, it is possible that the end concentration in the final product is lower.

Response: The values obtained according to the method used were related to the concentration of total fatty acids, as is the one available in our working place, moreover the results were obtained according that.

Regarding the flavour profiles, the authors mostly focus on volatiles related to lipid oxidation. In cooked meat, Maillard reaction products also played an important role in the flavour of the product. Did the authors also consider looking at such products?  

Response. The Maillard reaction products were not detected with the method used. Probably these compounds were under the detection limit. Next time we will try to optimize the method to detect the Maillard compounds. We might have had to cook the burger at higher temperature, use another extraction fibre with more absorbing capacity to those compounds, and higher extraction temperature. A suitable method for this purpose could be that published by He et al. (2021). Journal of Food Science, 86(2), 293.

The added value of the PCA analysis based on FA profile is not so clear. It would be important to highlight it better in the discussion of the results.

Other comments:

- Line 79: “from” is repeated twice. - Changed

- Section 2.1: The fatty acid profile of pork fat and olive oil is provided, while only the total SFA, MUFA and PUFA content is reported for sunflower oil. It would be nicer to be consistent and either report the fatty acid profile for all or only the total SFA/MUFA/PUFA for all.

- Lines 196-197: It is a bit misleading to first state that the pH values were similar for all treatments and then, mentioned that they were different.

Response: It was changed, mentioning that the difference through the three formulation is a small one.

- Line 199: It would be nice to mention here the water content of pork backfat between brackets to give a clearer idea on how much lower it is compared to oleogels.

Response: The authors considered that this piece should be changed, focusing in the addition of the water from the oleogels.

- Lines 201-203: The authors mentioned that the variations in water activity cannot be attributed to the oleogel mineral salts. Do they have an hypothesis on what could explain their outcomes? Also, although the water content in oleogels is higher than in pork fat, the moisture content in the burgers are similar. Can the authors comment on that?

Response 1. It has been commented that the lower water activity in the oleogel burgers could be attributed not only to ions but also to OH-groups of glucose. The changed text is the following:

“Oleogels contained 56% of water, which is a considerably higher amount than the water content of pork backfat and, therefore, it would have been expected that aw of oleogel burgers were higher than that of GPF ones. However, the aw of GPF burgers was the highest among the formulations tested. The aw-reducing effect of oleogels could be attributed to its components, namely mineral salts (CaSO4, Na2H2P2O7), glucose and Na+ from the alginate and ascorbate salts incorporated into the oleogel burgers. The OH-groups of glucose might have played a significant role on that effect (Zuorro et al., 2021) considering that the amounts of ash or Na+ originated from the oleogels were not high enough to cause a significant increase in ash and NCl contents of the oleogel burgers as regards to GPF burgers (Table 2).”

Response 2. The text has been changed to explain the lack of significant differences in moisture (see red coloured letters).

“Concerning the proximate composition, in agreement with burger formulations (Table 1)... Despite the water incorporated into the mix through the addition of oleogels (approximately 2.3%) resulted in higher moisture mean values, the variation in moisture content among batches might explain that moisture differences were not significant.”

- Line 212: Could the authors be more specific in why the use of olive oil changes the yellowness, for example by the presence of specific pigments in olive oil?

- Lines 217-218: Did the authors consider measuring the antioxidant capacity of their fat to confirm their hypothesis?

Response: To measure the antioxidant capacity of the fat would be a mean to strengthened the hypothesis. However, when the experiment was planned, we missed to consider this possibility. 

- Lines 226-227: Although the presence of collagen in pork fat could explain the higher content of collagen in the burger with pork fat compared to sunflower oil, it is unclear then why no such difference was observed between the burger with pork fat and the one with olive oil oleogel.

Response: This lack of difference can be due the heterogenicity of the product, considering that the goat meat has its content of collagen, the portions tested may be the ones that had a higher concentration of this compound.

- Line 229: major changes in the FA profile instead of major changes at the FA profile. - Changed

- Lines 260-262: It is unclear what the authors mean by cooking yield and where the reader can find the value related to the cooking yield.

Response: For easier comprehension it was changed for weight loss.

Response. Cooking yield has been replaced by weight losses to avoid confusion due to synonymy. The changes in the text has been marked in red letters.

“Weight loss percentage presented significant differences, with GSF being the treatment with the lowest weight losses. Previous studies using oleogels to replace animal fat in cooked burger have found that the oleogel effect on weight losses was variable depending on lipid source, emulsion and gelling or structuring agents.”

- Reporting the standard deviation (SD) for each burger rather than the overall SEM in tables and figures will provide the reader a better estimation of the variability of the results and help with their interpretation.

Response: It was considered that the use of the SEM would make it easier to evaluate and perceive the results obtained, by reducing the number of values exposed without compromising the comprehensiveness of the work.

- Line 287: The term “initial time” is confusing, maybe reformulate to refer to the time just after cooking. – Changed

- Line 293: “was slower” instead of “was more slowed down” - Changed

- Tables: It would be nice to precise in the footnotes that the values presented are means and that they were obtained on triplicates with a sentence such as “Values are means (n=3)”

- Figure 2: It would be good to precise in the caption what the bars and error bars mean, i.e. Means +/- SD? Also, this figure seems to have a light grey outline that the authors might want to remove. - Changed

- Line 355: major instead of mayor - Changed

- Table 6: The first column is titled “Column”, another name such as “Fatty acid” would be more appropriate. Also, in this column, it would make more sense to bold the fatty acid names that were significantly different (i.e. as in the F ratio and Prob>F columns) rather than all the other ones. - Changed

Reviewer 3 Report

Title: Use of olive and sunflower oil oleogels as pork fat replacers in goat meat burgers: Fat reduction and effects in lipidic quality

This study was evaluate the quality of goat meat burgers made with olive oil and sunflower oil-based oleogels, and the physicochemical pH, water activity, composition, fatty acid profile, color, yield, texture, oxidative  stability and volatile compounds were analyzed systematically. The study was generally interesting, as oleogels was investigated in the food meat system. However, the manuscript was properly deficit, especially the introduction and methods section. Major revision is recommended.

Below are some comments and questions required to be addressed:

Q1. L19, oleo gels→oleogels

Q2. L79, remove a “from”

Q3. The fatty acid profile of the pork fat, olive oil and sunflower oil used in the work should be measured rather than quoted.

Q4. Provide the preparation information of oleogels.

Q5. L175, TBARS contents were determined following de above-mentioned procedure?? Provide detailed formation.

Q6. Provide detailed formation about the analyze of Volatile compounds.

Q7. vegetable oil FA?

Q8. L367-369 “By performing a stepwise discriminant analysis, it is possible to identify and correctly classify 100% of the burgers using only two variables corresponding to the FA, C20:2n-6 and C18:1n-9, and the model is highly significant (p < 0.0001*; Wilks Lambda value: 0.00475) which is possible to see in table 6.” Why is the author to identify and correctly classify 100% of the burgers? In this case, there are only three samples used. This is insufficient.

Q9. In the Introduction, the author should focus on the analysis of the current state abot the research and reinforced the highlights.

Author Response

Dear reviewer,

All modifications were made following the reviewer's suggestions and comments, and responses to their comments are also attached. Thanks to their recommendations, significant modifications were made throughout the manuscript.

Thank you for your attention.

This study was evaluate the quality of goat meat burgers made with olive oil and sunflower oil-based oleogels, and the physicochemical pH, water activity, composition, fatty acid profile, color, yield, texture, oxidative  stability and volatile compounds were analyzed systematically. The study was generally interesting, as oleogels was investigated in the food meat system. However, the manuscript was properly deficit, especially the introduction and methods section. Major revision is recommended.

Below are some comments and questions required to be addressed:

Q1. L19, oleo gels→oleogels - Changed

Q2. L79, remove a “from” - Changed

Q3. The fatty acid profile of the pork fat, olive oil and sunflower oil used in the work should be measured rather than quoted.

Response: As there are several works regarding pork fat and olive oil used, the authors decided to use previous studies. Relating to the sunflower oil profile, by the time of the experimental procedures the company was contacted but there were no responses, so the brand allegations were used.

Q4. Provide the preparation information of oleogels.

Response: The oleogels were prepared according to the reference mentioned in the methods session, in order to not add unnecessary text that can be reached easily, the authors considered that describing the oleogels preparation would not be needed.

Q5. L175, TBARS contents were determined following de above-mentioned procedure?? Provide detailed formation.

Response. The reference for the method has been provided at the end of the sentence. See letters in red.

“TBARS contents were determined following de above-mentioned modified NP-ISO-3356/2009 [33] procedure as described for the raw burger TBARS determination.”

Q6. Provide detailed formation about the analyze of Volatile compounds.

Response. Further details for the analysis have been provided (see red letters).

Volatile compounds were analyzed by gas chromatography-mass spectrometry using the solid-phase micro-extraction (SPME) technique following the procedure described by Carballo et al. [47] with the following modifications. Briefly, the extraction of volatiles was carried out with a CTC Pal automated system (Agilent Technologies; Santa Clara, CA, USA) equipped with an automatic SPME injection device using a 30-min 250-ºC conditioned 75 mm Carboxen/polydimethylsiloxane-1-cm-coated fused silica fibre, from 3 g of grounded burger sample placed into a 20-ml screw cap vials, which were previously incubated at 45 ºC during 20 min, for a 40 min exposition period at 40 ºC. A 60 m x 0.25 mm sized, 0.25 mm film thickness DB-5MS column (J&W Scientific, Folson, CA, USA) and helium (1 ml/min) were used for separation with the temperature being programmed at 35 ºC (1 min), 35 ºC to 50 ºC (10 ºC/min), 50 ºC to 200 ºC (4 ºC/min), 200 ºC to 250 ºC (50 ºC/min) and 250 ºC (11 min). The mass spectrometer transfer line and ion source temperatures were 240 º and 260 ºC, respectively, and the detector operated in electron impact mode (70 eV) scanning from 40 m/z to 350 m/z. Identification was carried out by comparing the mass spectra with those contained in the NIST/EPA/NIH-98 mass spectral database and the linear retention indexes, experimentally calculated using a series of n-alkanes with those reported in the literature [Carballo et al. 2020; 47]. The concentrations of the identified compounds were expressed as area units (AU) x 106.

Q7. vegetable oil FA?

Response: This phrase relates to the fatty acid profile from the vegetable oil used.

Q8. L367-369 “By performing a stepwise discriminant analysis, it is possible to identify and correctly classify 100% of the burgers using only two variables corresponding to the FA, C20:2n-6 and C18:1n-9, and the model is highly significant (p < 0.0001*; Wilks Lambda value: 0.00475) which is possible to see in table 6.” Why is the author to identify and correctly classify 100% of the burgers? In this case, there are only three samples used. This is insufficient.

Response: There are 3 formulations, and from these, replicates were evaluated, so the final number of samples was 18, as is possible to observe in figure 2 there are 6 dots for each cluster.

Q9. In the Introduction, the author should focus on the analysis of the current state abot the research and reinforced the highlights.

Response: Having in mind that the studies mentioned as references for the work, and that previous studies with burgers did not contemplate specifically goat meat burgers with fat replacement, the authors intended to focus on the motives for the goat meat burgers to be studied and why the proposed replacement was considered.

Round 2

Reviewer 1 Report

The main question, which is the significance of this research, I asked has not been well addressed. Please give solid evidence to prove that lard has been widely used in meat products. As I know, lard itself has health concerns that would prevent it from being a fat source for other foods. The only answer from the authors' narrative "normally the fat source for meat products is pork fat", which is inconsistent with my knowledge. Please give examples of the application of lard in other foods, especially goat meat products.
In addition, the question "Please list all the ANOVA results in the figures and tables." has been ignored.

Author Response

Dear reviewer,

We thank you again for the time spent reviewing the article. Alterations were made in order to improve the article and answer the question addressed. Considering the recommendations, more modifications were made throughout the manuscript.

Thank you for your attention.

____________________________________________________________________

*The main question, which is the significance of this research, I asked has not been well addressed. Please give solid evidence to prove that lard has been widely used in meat products. As I know, lard itself has health concerns that would prevent it from being a fat source for other foods. The only answer from the authors' narrative "normally the fat source for meat products is pork fat", which is inconsistent with my knowledge. Please give examples of the application of lard in other foods, especially goat meat products.

Response: We have documented with two citations the fact that pork fat is commonly used to prepare ruminant (goat) meat products and we have focused the statement on occidental countries or countries with a frequent pork-eating culture. The reference by Feiner, in a relevant book on meat technology, include the following sentence in the chapter 27 on burgers and patties “Burgers made from mutton commonly contain 20–25% fat as well as spices, water and other flavours.” (p 487)”. The other reference is the published proceeding IOP Conference Series: Earth and Environmental Science by Stajic and Pisinow (2021) entitled goat meat products who stated in the abstract that “The negative impact of goat meat on the properties of meat products is mainly associated with the use of goat fatty tissue. However, this could be overcome by using fatty tissue of other animals (e.g. pork back fat or beef fatty tissues)” and then in the text repeat and explain that “Also, Leite et al. [7] reported that fresh sausages prepared with culled goat meat and 30% of pork fat had good acceptability” and similar comments can be found in the text for cooked products or dry fermented sausages. Accordingly, we have done the following changes in the text: 

L44-6. Goat meat is less popular by far than pork, beef or chicken in the occidental countries daily diet. However, goat meat is part of local food traditions, goat meat eaters represent a niche market, and goat meat consumption appeared to show a trend to increase in the last decade [13,14]. 

L55-57. In the countries where the pork consumption is extended, the usual source of fat used to prepare fat-containing ruminant meat products, including goat meat products, such as sausages or patties is pork fat (Feiner et al., 2006 [19]; Stajic and Pisinov, 2021 [20]), due to its favorable technological properties and positive effects on their eating quality. However, its lipidic profile is not desirable when concerning about healthy food [3-7]. 

*In addition, the question "Please list all the ANOVA results in the figures and tables." has been ignored.

Response: We apologize for the mistake, tables were changed and the values were included.

Reviewer 3 Report

I recommend the author “The fatty acid profile of the pork fat, olive oil and sunflower oil used in the work should be measured rather than quoted.”,because of the fatty acid composition of different parts of lard varies greatly as well as  olive oil and sunflower oil  from different varieties and regions.

In the Materials and Methods, the analysis of Volatile compounds still can't get.

After perfecting these, think it can be publicated in Biomolecules

Author Response

Dear reviewer,

Modifications were made following the reviewer's suggestions, and responses to the comments are also attached. Thanks to their recommendations, more modifications were made throughout the manuscript.

Thank you for your attention.

______________________________________________________________________

*I recommend the author “The fatty acid profile of the pork fat, olive oil and sunflower oil used in the work should be measured rather than quoted.”,because of the fatty acid composition of different parts of lard varies greatly as well as  olive oil and sunflower oil  from different varieties and regions.

Response: We appreciate your recommendation. Our main problem in this respect is that we did not analyse the sunflower oil FA composition and we have not more information than that from the label. We think It is not worthy to analyse the FA in a recent purchase oil bottle from the same brand because the FA profile would be different due to time-related batch-to-batch variation. The other two sources or lipids in addition to be quoted can be found tabulated in other articles.

*In the Materials and Methods, the analysis of Volatile compounds still can't get.

Response: We forgot to add more info on the procedure for volatile analysis. Sorry for the inconvenience. In this version it has been included.
